# A systems biology analysis of lipolysis and fatty acid release from adipocytes *in vitro* and from adipose tissue *in vivo*

William Lövfors[1,2], Jona Ekström[1], Cecilia Jönsson[3], Peter Strålfors[3], Gunnar Cedersund[1,4], Elin Nyman[1]*

**1** Department of Biomedical Engineering, Linköping University, Linköping, Sweden, **2** Department of Mathematics, Linköping University, Linköping, Sweden, **3** Department of Biomedical and Clinical Sciences, Linköping University, Linköping, Sweden, **4** Center for Medical Image Science and Visualization (CMIV), Linköping University, Linköping, Sweden

* elin.nyman@liu.se

**Data Availability Statement:** The experimental data as well as the complete code for data analysis and modelling are available at https://github.com/willov/lipolysis (DOI: 10.5281/zenodo.5070401)

## Abstract

Lipolysis and the release of fatty acids to supply energy fuel to other organs, such as between meals, during exercise, and starvation, are fundamental functions of the adipose tissue. The intracellular lipolytic pathway in adipocytes is activated by adrenaline and nor-adrenaline, and inhibited by insulin. Circulating fatty acids are elevated in type 2 diabetic individuals. The mechanisms behind this elevation are not fully known, and to increase the knowledge a link between the systemic circulation and intracellular lipolysis is key. However, data on lipolysis and knowledge from *in vitro* systems have not been linked to corresponding *in vivo* data and knowledge *in vivo*. Here, we use mathematical modelling to provide such a link. We examine mechanisms of insulin action by combining *in vivo* and *in vitro* data into an integrated mathematical model that can explain all data. Furthermore, the model can describe independent data not used for training the model. We show the usefulness of the model by simulating new and more challenging experimental setups *in silico*, e.g. the extra-cellular concentration of fatty acids during an insulin clamp, and the difference in such simulations between individuals with and without type 2 diabetes. Our work provides a new platform for model-based analysis of adipose tissue lipolysis, under both non-diabetic and type 2 diabetic conditions.

## Introduction

The combination of experiments and mathematical modelling to understand the human body is an old idea, dating back to at least Norbert Wiener and the introduction of cybernetics [1]. One of the most well-developed such endeavours concerns modelling of metabolism, which has been an active field since the 1960s [2–4]. Lately, the idea has gained new popularity with the concept of a digital twin, which describes the physiology and biochemistry of a patient using mathematical computer models [5–9]. In practice, such a digital twin is produced by the

and is mirrored at https://gitlab.liu.se/ISBgroup/projects/lipolysis.

**Funding:** PS acknowledges support from Linköping University, the Swedish Diabetes Fund (a 3-years program; https://www.diabetes.se/diabetesfonden/), and the Swedish Research Council (a 5-years program; https://www.vr.se/). EN acknowledges support from the Swedish Research Council (Dnr 2019-03767), the Heart and Lung Foundation (https://www.hjart-lungfonden.se/), CENIIT (20.08; http://ceniit.lith.liu.se/en/), and Åke Wibergs Stiftelse (M19-0449; https://ake-wiberg.se/). GC acknowledges support from the Swedish Research Council (Dnr 2018-05418, 2018-03319), Swedish Foundation for Strategic Research (ITM17-0245; https://strategiska.se/), SciLifeLab and KAW (2020.0182; https://www.scilifelab.se/), Horizon 2020 (PRECISE4Q, 777107; https://ec.europa.eu/programmes/horizon2020/), CENIIT (15.09), ELLIIT (https://www.lu.se/forskning/starka-forskningsmiljoer/strategiska-forskningsomraden/elliit), and the Swedish Fund for Research without Animal Experiments (https://forskautandjurforsok.se/swedish-fund-for-research-without-animal-experiments/). The funders had no role in study design, data collection and analysis, decision to publish, or preparation of the manuscript.

**Competing interests:** The authors have declared that no competing interests exist.

incremental integration of partial insights and data [10–12]. In this paper, we produce one new such integration: that of lipolysis studied *in vitro* and *in vivo*.

Lipolysis, the breakdown of triacylglycerol to glycerol and fatty acids, and the subsequent release of fatty acids and glycerol as energy fuel for other organs, is one of the main functions of the adipose tissue. Because of the critical role of fatty acids as a fuel for the body, this function is also central to energy homeostasis. Interest in lipolysis has gained more traction as the prevalence of obesity, type 2 diabetes and its sequelae have increased dramatically over the last decades. Lipolysis is stimulated in the body mainly by the catecholamine noradrenaline, which is released locally in the adipose tissue, and by adrenaline in the circulation. The two catecholamines signal through $\alpha_2$- and $\beta$-adrenergic receptors stimulate lipolysis by increasing intracellular levels of cyclic AMP (cAMP). An increased concentration of cAMP results in the activation of adipose triacylglycerol lipase (ATGL) and hormone sensitive lipase (HSL), the two rate-limiting lipases responsible for lipolysis. Insulin counteracts the stimulation of lipolysis in adipocytes by activation of phosphodiesterase 3B (PDE3B) that degrades cAMP and thereby reduces the rate of lipolysis [13–15]. The two catecholamines can also inhibit lipolysis by inhibiting the activation of adenylate cyclase through the $\alpha_2$-adrenergic receptor. Lipolysis is thus under tight positive and negative hormonal control.

The signalling pathways involved in the control of lipolysis are highly complex, and numerous crosstalks between different pathways and branches are emerging. Jönsson *et al.* [15] provide detailed elucidation of the pathways controlling lipolysis in adipocytes from human subcutaneous adipose tissue and show a new $\beta$-adrenergic—insulin crosstalk, where $\beta$-adrenergic signalling, in addition to stimulation, also inhibits lipolysis via parts of the insulin signalling pathway. The signalling pathways also include an additional stimulatory lipolytic action of insulin at high concentrations. Beyond the actions mentioned in [15], Stich *et al.* also suggest an anti-lipolytic action of insulin involving $\alpha_2$-adrenergic receptors [16]. This action was observed during microdialysis experiments, *in situ* in human subcutaneous adipose tissue, stimulated with a protocol involving adrenaline, isoproterenol, insulin and phentolamine. The high degree of crosstalk and the different actions at different concentrations of the hormones controlling lipolysis make it hard to successfully grapple with experimental data by mere reasoning. To understand the role and relative importance of these different actions of insulin and the catecholamines in the control of lipolysis, a next step therefore is to test the suggested mechanisms in a formalized way using mathematical modelling.

We have earlier, in several steps, developed mathematical models for insulin signalling in human adipocytes: first in isolation and later connected to models of systemic glucose control, and used the models to unravel key alterations in type 2 diabetes [17–20]. These models, however, do not include lipolysis and the control of lipolysis by insulin. We have also studied systemic whole-body effects of fatty acids on glucose uptake and release, using modelling [21], but with no link to intracellular lipolysis. For a more thorough review of models in diabetes, we refer to [22]. There have also been other efforts to understand adipose tissue lipolysis in more detail, for example experimentally in [23] and using mathematical modelling in [24, 25], but without detailed intracellular components. In summary, none of the existing models have been developed to elucidate the mechanisms of control of intracellular lipolysis.

Here, we develop a new minimal model for lipolysis and the release of fatty acids based on both *in vitro* and *in vivo* experimental data from humans (Fig 1). The model includes all three suggested insulin actions to control lipolysis: two direct actions, one anti-lipolytic via protein kinase B (PKB) activation of phosphodiesterase 3B (PDE3B) (action-1), one lipolytic via inhibition of PDE3B (action 2), and a third indirect anti-lipolytic action via $\alpha$-adrenergic receptors (action-3). Using mechanistic modelling, we can evaluate the impact of these actions

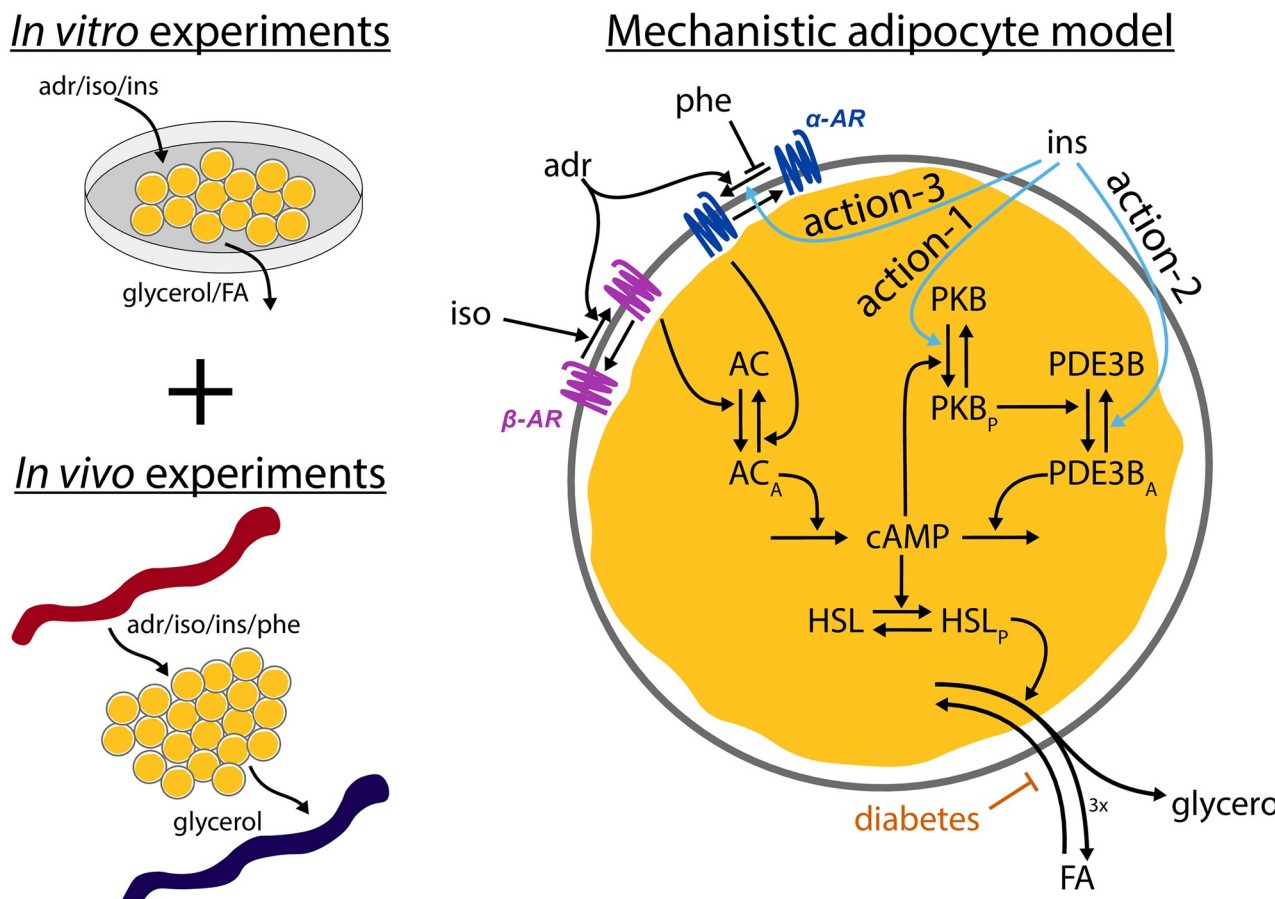

**Fig 1. The system under study.** Data from both *in vitro* and *in vivo* experiments of lipolysis control by insulin, adrenergic stimulus, and phentolamine were combined to create a first mechanistic model of lipolysis, under both non-diabetic and type 2 diabetic conditions. The model responds to stimuli with adrenaline (adr), isoproterenol (iso), insulin (ins), and phentolamine (phe), initiating signalling cascades through key proteins leading to release of fatty acids (FA) and glycerol. Adrenaline affects both β-adrenergic receptors (β-AR) and α-adrenergic receptors (α-AR), while iso only affects β-AR. Ins gives rise to three different insulin actions: action-1) an anti-lipolytic effect of insulin via protein kinase B (PKB) and phosphodiesterase 3B (PDE3B), action-2) a positive lipolytic effect via PDE3B at high insulin concentrations, and action-3) an anti-lipolytic effect of insulin via α-adrenergic receptors.

individually. The model accurately predicts independent validation data and is therefore useful to simulate new *in silico* experiments, such as the release of fatty acids *in vivo*, under both non-diabetic and type 2 diabetic conditions. The developed model is, to the best of our knowledge, the first model for the hormonal control of lipolysis, and it opens for new research and drug discovery related to type 2 diabetes.

## Results

To connect data from several sources in a common framework, we use mechanistic modelling. In mechanistic modelling, available knowledge about a system is formulated as a model by constructing a set of ordinary differential equations. The validity of such models can be tested by comparing model simulations to experimental data. Typically, the values of the model parameters, e.g. kinetic rate constants and initial concentrations of substances, are unknown and need to be estimated by training the model to experimental data. Other experimental data are then used for validating the predictive power of the model.

## Experimental observations and model development

We developed a mechanistic model focused on the regulation of intracellular lipolysis and the release of fatty acids and glycerol from the adipose tissue. The model is based on data from both *in vitro* measurements on isolated adipocytes, and *in vivo* microdialysis measurements, in both cases from non-diabetic individuals. More specifically, to develop the model we used experimental data from two sources: i) isolated adipocytes treated *in vitro* with the β-adrenergic agonist isoproterenol to stimulate lipolysis, and additionally with insulin to inhibit the isoproterenol-stimulated lipolysis, from [15], and ii) microdialysis measurements of lipolysis *in vivo*, stimulated with adrenaline/isoproterenol and inhibited by insulin and phentolamine, from [16]. Details on how the data was processed are described in the **Methods—Data processing** section.

We have taken three previously suggested mechanisms of crosstalk for the actions of insulin to explain the observed behaviour in the experimental data: an anti-lipolytic effect of insulin via protein kinase B (PKB) and PDE3B [15, 26]—action-1; a positive lipolytic effect of insulin via PDE3B at high concentrations of insulin [15]—action-2; and an anti-lipolytic effect of insulin via α-adrenergic receptors [16]—action-3. We have also included other known signalling steps in the control of lipolysis in adipocytes as indicated in Fig 1 and detailed in the Methods section. To avoid overfitting, the model was kept "minimal" in the sense that we focused on a few key proteins, and not every protein known to be involved in the control of lipolysis.

To further support the claim that the model is minimal, we performed a parameter identifiability analysis as detailed in the **Methods—Uncertainty estimation** section. In short, we estimated the minimal and maximal value any parameter could take while still yielding a statistically sufficient agreement between the model and the data. Parameters with bounded minimal and maximal values are identifiable, and parameters with unbounded values are non-identifiable. For computational reasons, we deemed a parameter non-identifiable if the value exceeded a given threshold (S1 Table). Any rate-determining parameter ($k_x$) that appeared to be downwards non-identifiable (i.e., approaching zero) was removed from the model, with the exception of the parameter determining the reesterification ($k8c$). In other words, no downwards non-identifiable parameters except for $k8c$ are present in the presented minimal model (Fig 1 and model equations). The reesterification parameter $k8c$ was not removed because it was necessary to later implement the effect of the diabetic condition, which was implemented as a reduction in reesterification. Thus, we would not be able to implement the diabetic condition without the reesterification. The parameter uncertainty bounds for the parameters in the minimal model is shown in Fig 2 and S1 Table. In total, 23 parameters were found to be identifiable.

## Comparisons between model simulations and data

The model was trained to *in vitro* dose-response data for the phosphorylation of PKB at Ser-473 and the release of glycerol and fatty acids in response to isoproterenol and insulin stimulation, as well as *in vivo* microdialysis data of glycerol release in response to adrenaline and insulin (Fig 3; solid lines represent the model simulation with the best agreement to data, shaded areas represent the model uncertainty, and experimental data are represented as mean values with error bars (SEM)). Here, the model uncertainty refers to the most extreme simulations, while still requiring the model to pass a $\chi^2$-test. The best model simulation clearly has a good agreement with the experimental data (Fig 3). This visual assessment is supported by a statistical $\chi^2$-test, where the cost of the model ($v^*$ for the optimal cost, see Methods), given the optimal parameter values ($\theta^*$, see S2 Table), is below the threshold of rejection given by the $\chi^2$-test ($v^* = 130.8 < \chi^2(0.05, 137) = 165.3$). The optimal parameter values are shown in S2 Table and in the set of scripts used to reproduce the results (see **Data and model availability**). The model

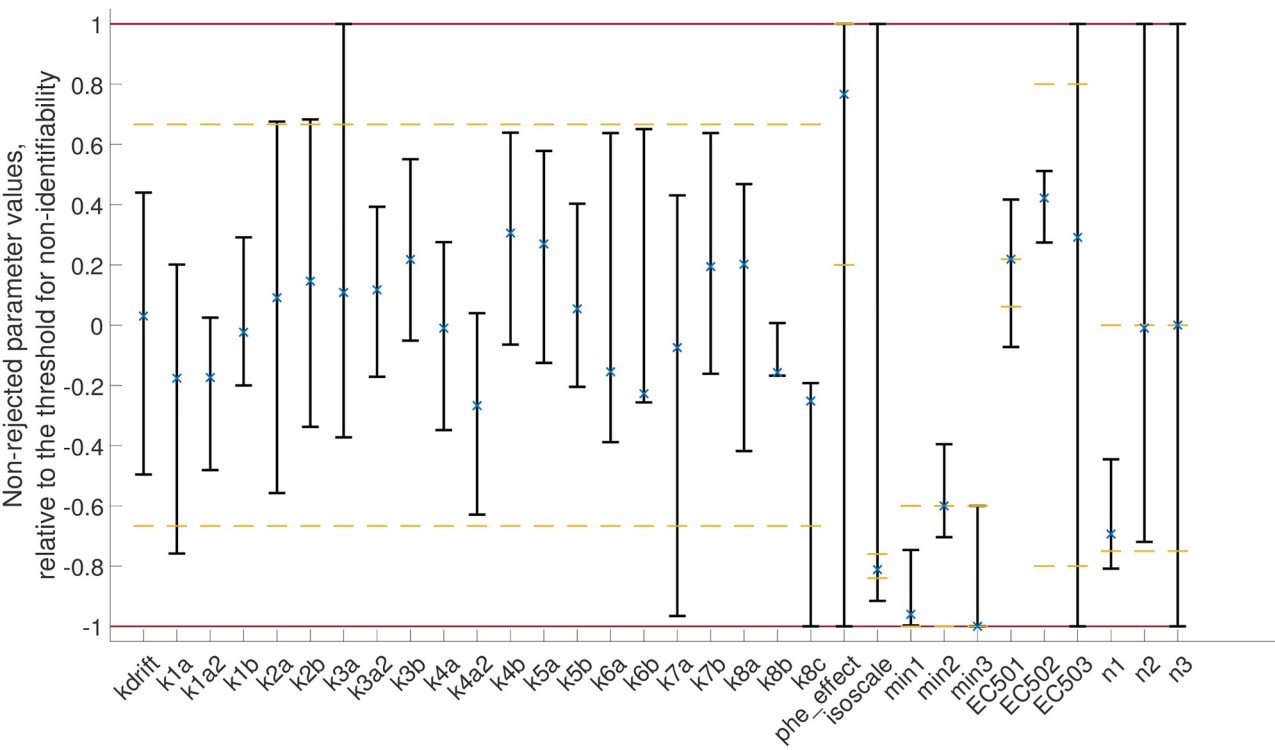

**Fig 2. Parameter identifiability analysis.** The minimal and maximal values of a parameter was found using the optimization approach detailed in the **Methods—Uncertainty estimation** section. The parameter values are expressed as relative values with respect to threshold for non-identifiability, where -1 represents a parameter value at the lower threshold, and 1 represents a parameter at the upper threshold. x represent the optimal parameter values and yellow dashed lines represent the bounds for a specific parameter when estimating parameter values. The thresholds for identifiability, and maximal and minimal values for the parameter values are given in S1 Table. The bounds on the parameter values and the values of the optimal parameter found during in the parameter estimation is given in S2 Table.

uncertainty was estimated in the same way as in [27], by maximizing/minimizing the simulation in all experimental data points while requiring the cost to not exceed the $\chi^2$-threshold.

## Model validation: Predicting intracellular phosphorylation of HSL

For a model to be of practical use, it should be able to perform reasonable predictions. To test this, we used the model to predict the dose-response for phosphorylation of HSL (HSLp), an intracellular state in the model (Fig 1) that was not used when training the model to data. We estimated the uncertainty of the model prediction in the same way as described in the comparison between model and data, i.e. we maximized/minimized the prediction simulation, while requiring the agreement to the estimation data to be acceptable (i.e., keeping the cost below the threshold, see **Statistical analysis** in Methods). When compared to the experimental data (Fig 4), the model prediction with the best agreement to the new data overlaps well. This agreement is statistically supported using a $\chi^2$-test ($v^* = 10.7 < \chi^2(0.05, 10) = 18.3$). The model prediction uncertainty is shown in S2 Fig.

## Investigating the different actions of insulin

With the validated model, we continued to investigate the impact of the three different insulin actions (Fig 1, the three blue arrows) by excluding one action at a time. We excluded an action by keeping the corresponding $Ins_x$ variable (see Eq (1)) at basal levels throughout the

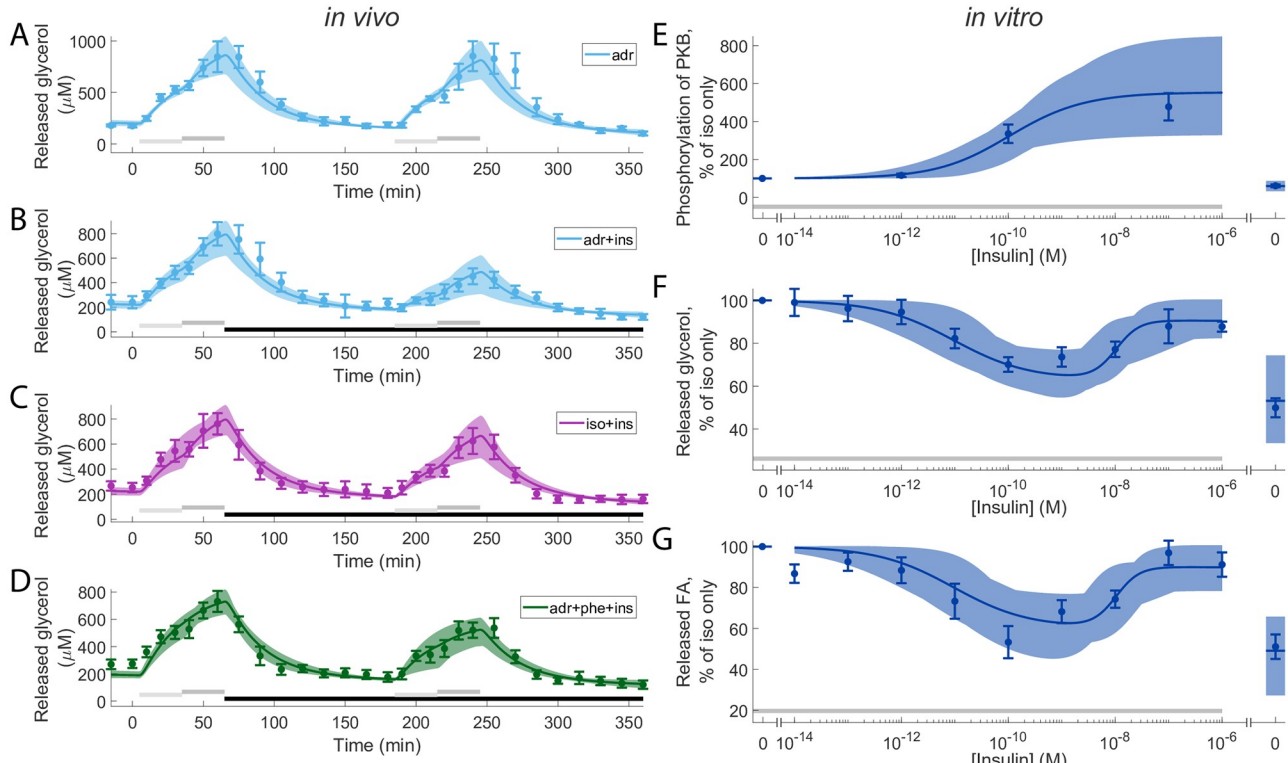

**Fig 3. Model agreement with experimental data.** In all panels, solid lines represent the model simulation with the best agreement to data, the shaded areas represent the model uncertainty, and experimental data points are represented as mean values with error bars (SEM). (A-D), *in vivo* time-series experiments. (E-G),*in vitro* dose-response experiments. In all subfigures, horizontal bars indicate where stimulations were given. In detail, light/dark grey bars indicate stimulation with: 1/10 μM, respectively, adrenaline in (A,B), 0.1/1 μM isoproterenol in (C), and 1/10 μM adrenaline with 100 μM phentolamine. Black bars in (B-D) indicates stimulation with 0.6 nM insulin. In (E-G) grey bars indicate stimulation with isoproterenol (10 nM). In the *in vivo* experiments, experiments with adrenaline are shown in light blue (A-C), with isoproterenol in purple (B), and with the combined stimulation with adrenaline and phentolamine in green (C). In the *in vitro* experiments (D-F), increasing doses of insulin were given together with 10 nM isoproterenol in all points except one. The point without isoproterenol got no stimulus and is shown to the right in the graphs. An alternative visualization is available in S1 Fig showing the difference by overlaying the experiments.

simulation (instead of increasing with increased concentration of insulin). Firstly, by removing action-1 (the anti-lipolytic effect of insulin via PKB-mediated activation of PDE3B), the model is unable to explain the decline in glycerol release in response to increased levels of insulin *in vitro*: compare Fig 5A with Fig 5C. Secondly, by removing action-2 (the positive lipolytic effect of insulin via inhibition of PDE3B), the model is unable to explain the recovery in glycerol release at high insulin concentrations *in vitro*: compare Fig 5A with Fig 5E. Finally, by removing action-3 (the anti-lipolytic effect of insulin via α-adrenergic receptors), the model is unable to explain the decrease in glycerol release in the second set of adrenergic stimuli *in vivo*: compare Fig 5B with Fig 5D. The removal of insulin action-1 and -2 renders the model unable to agree with the experimental data sufficiently well ($v_1^* = 501.1 > \chi^2(0.05, 137) = 165.3$ and $v_2^* = 181.8 > \chi^2(0.05, 137) = 165.3$ for the removal of action-1 and -2 respectively). With the removal of action-3 the model can still quantitatively explain the data sufficiently well ($v_3^* = 126.4 < \chi^2(0.05, 137) = 165.3$), but not qualitatively.

The reason why the model is not qualitatively good enough without insulin action-3 can be seen in Fig 5D. In the data from the experiment without phentolamine (blue error bars) insulin inhibits the release of glycerol during the second set of stimuli with adrenaline (at 190 to 240 minutes) relative to the first set of stimuli with adrenaline without insulin stimulation (at 10 to

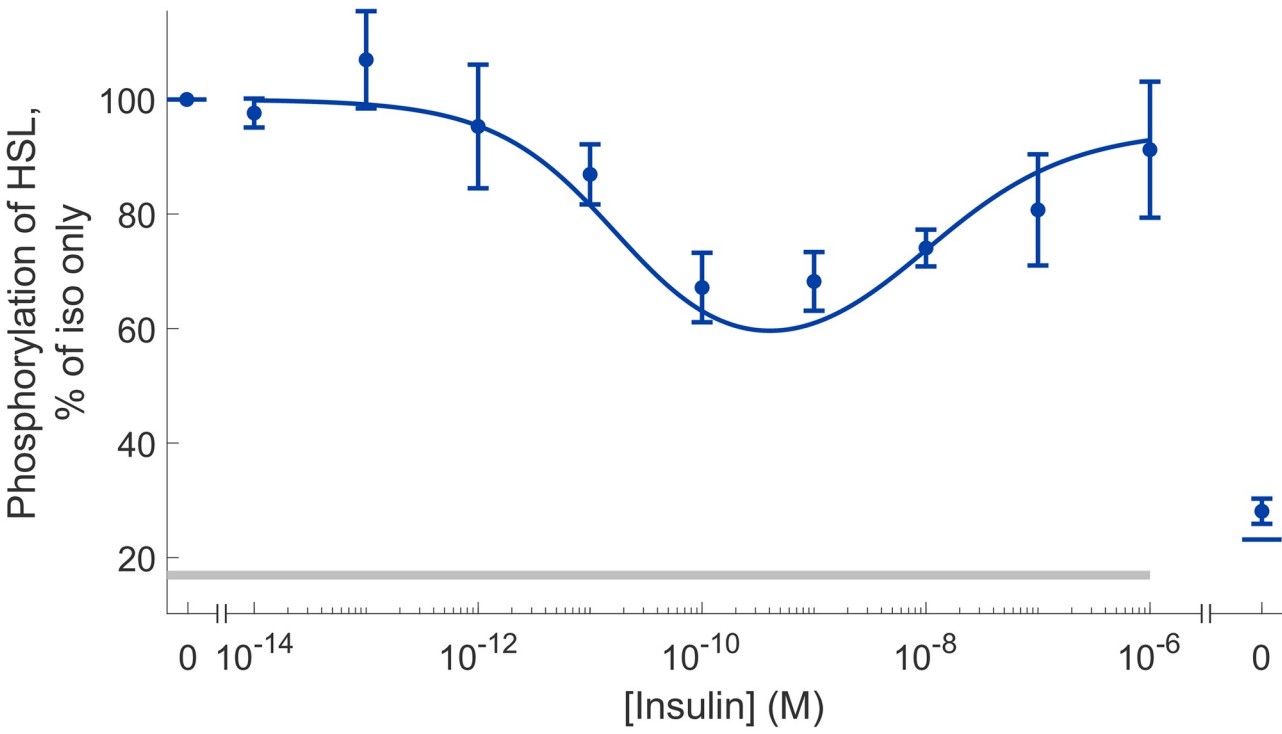

**Fig 4. Prediction of intracellular extent of HSL phosphorylation.** The line represent the model prediction with the best agreement with the validation data, and the experimental data are represented as mean values with error bars (SEM). The horizontal grey bar indicates where stimulation with isoproterenol (10 nM) have been given. Increasing doses of insulin were given together with 10 nM isoproterenol in all points except one. The point without isoproterenol got no stimulus and is shown to the right in the graph. The full model prediction uncertainty is shown in S2 Fig.

60 minutes). This inhibitory effect by insulin on the release of glycerol during the second set of stimulation with adrenaline is blunted in the data from the experiment with phentolamine (green error bars). Furthermore, the effect of phentolamine on the release of glycerol during the first set of stimuli with adrenaline is markedly lower than when insulin is added in the second set of stimuli. In other words, the effect of phentolamine on the release of glycerol is in a sense insulin dependent. This behaviour of phentolamine having an insulin dependent effect is not exhibited by the model when insulin action-3 is removed. In Fig 5D, the effect of adding phentolamine can seen by comparing the simulation with phentolamine (green line) with the simulation without phentolamine (blue line). The effect of phentolamine is essentially the same in both sets of stimuli with adrenaline. In other words, the effect of phentolamine is not insulin dependent in the model simulations. Therefore, the model without insulin action-3 is not qualitatively good enough. The best agreement between the model without insulin action-3 and all experimental data is shown in S3 Fig. Consequently, all three actions of insulin are required for the model to explain the available experimental data.

## Estimating the extent of altered reesterification in type 2 diabetes

We then used the validated model to gain new biological insights. As demonstrated in [15, Fig 14] at the cellular level, essentially only the release of fatty acids, but not of glycerol and hence not lipolysis, is affected in type 2 diabetes. The authors conclude that this is due to reduced reesterification, i.e. a decreased reuse of fatty acids to re-form triacylglycerol, in the diabetic state. We therefore added a single parameter representing a decrease in reesterification to the model to represent the type 2 diabetic condition (*diab* in Eq (9)). In addition to extending the

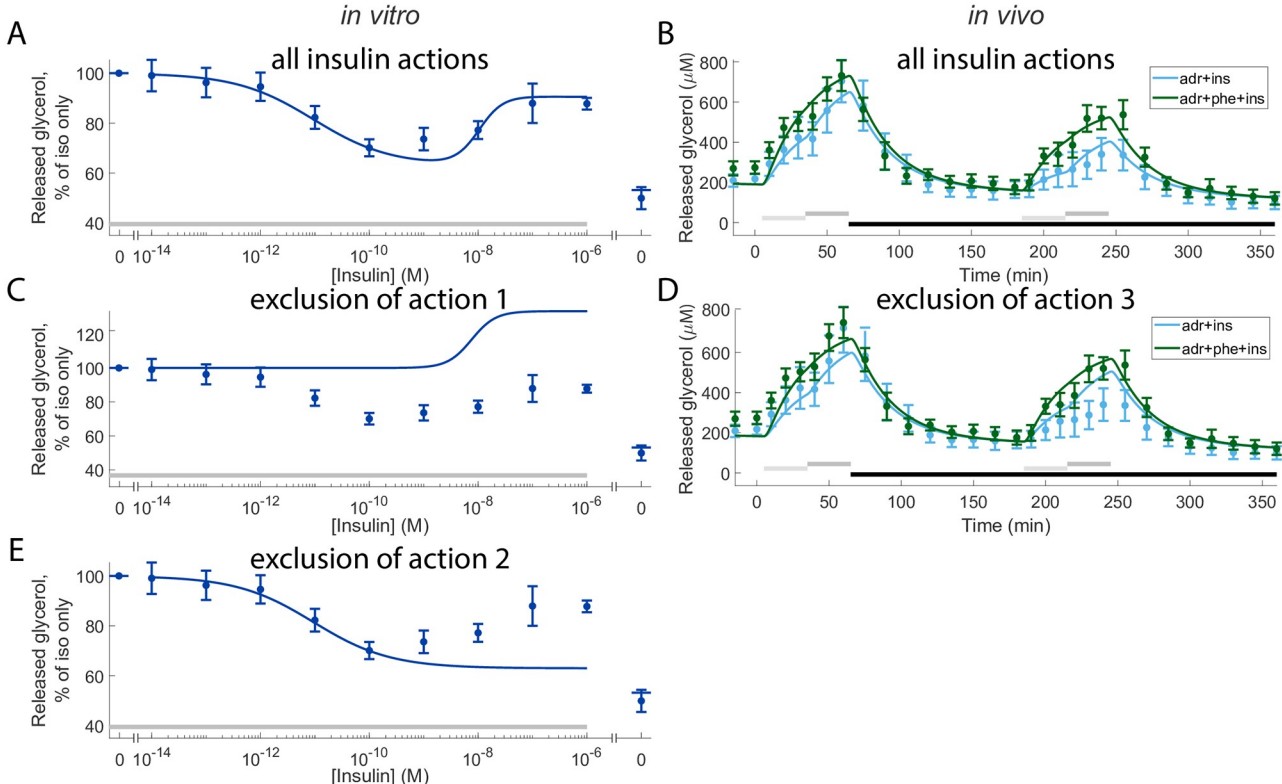

**Fig 5. Effects of excluding either of the three insulin actions.** In panels (A-E), data points with error bars represent mean and SEM values and solid lines represent the model simulation with the best agreement to data. (A, B), model simulations with all insulin actions present (same as Fig 3C and 3F), see Fig 1 for a graphical representation of the three insulin actions. (C-E), the model simulations when either of the actions are excluded. In all subfigures, horizontal bars indicate where stimulations were given. In (A, C, E) grey bars indicate stimulation with isoproterenol (10 nM) and in (B, D) light/dark grey bars indicate stimulation with low/high dose of adrenaline (1/10 μM) with or without phentolamine (100 μM), and black bars indicate stimulation with insulin (0.6 nM). In the *in vitro* experiments (A, C, E), increasing doses of insulin were given together with 10nM isoproterenol in all points except one. The point without isoproterenol got no stimulus and is shown to the right in the graphs. In the *in vivo* experiments, experiments with adrenaline are shown in light blue (B, D) and with the combined stimulation with adrenaline and phentolamine in green (D).

model, we also extended the set of experimental data beyond the data used so far (see e.g Fig 3). The set of experimental data now also includes the phosphorylation of HSL previously used for validation (Fig 4), as well as reesterification under type 2 diabetic conditions ([15, Fig 14F]). We then trained the extended model with the extended set of experimental data and quantified the maximal range of reesterification under both normal and type 2 diabetic conditions. The model agrees well with the experimental data (Fig 6 and S4 Fig), supported by a $\chi^2$-test ($v^* = 164.1 < \chi^2(0.05, 152) = 181.8$), and accurately shows that only the release of fatty acids and thus reesterification is affected under type 2 diabetic conditions (Fig 6). With the trained model, we found the range of reesterification to be altered from 66.7—74.3% under normal conditions to 39.6—64.1% under diabetic conditions.

## *In vivo* model simulations of fatty acid release, under both non-diabetic and type 2 diabetic conditions

In addition to predicting dose-response data or quantifying the range of impairment of the reesterification of fatty acids, we can also use the model to predict temporal changes *in vivo*. During lipolysis both glycerol and fatty acids are released from the adipocytes (Fig 1). However, only the time series for glycerol release were measured in the *in vivo* data used to train

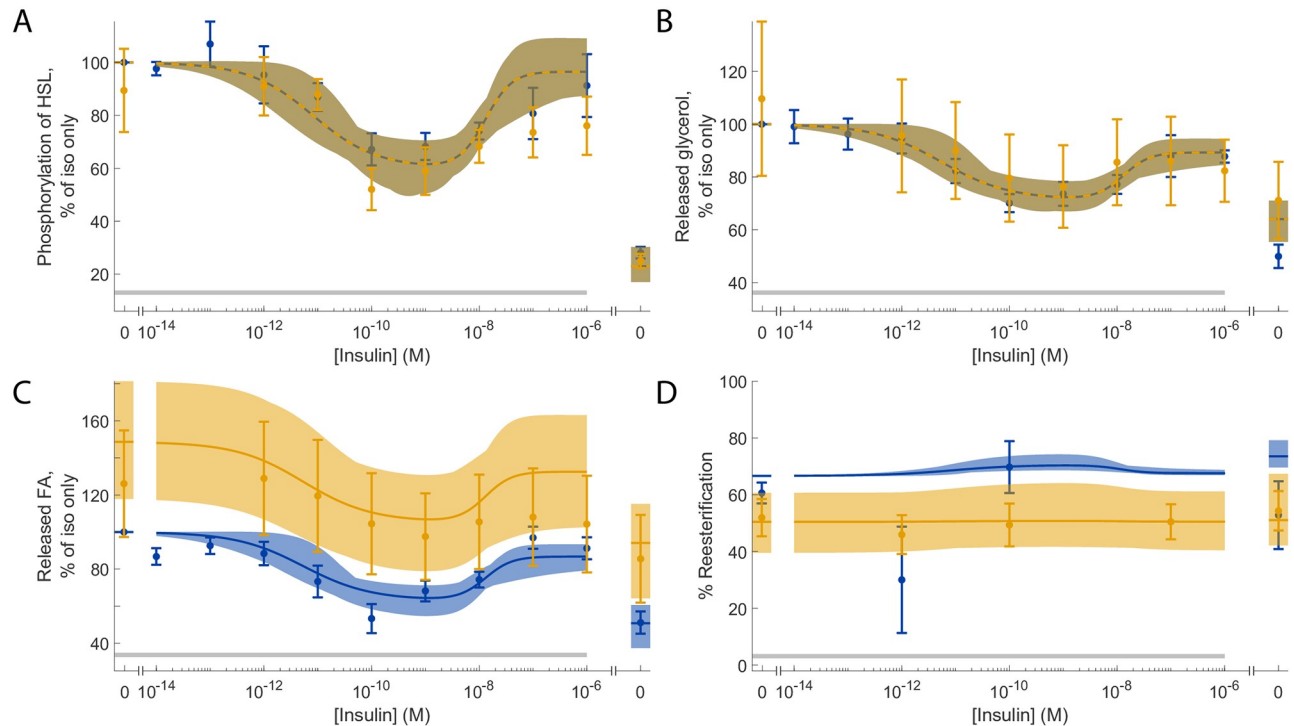

**Fig 6. Model agreement when trained to the extended set of experimental data.** In all panels, dots and error bars represent mean and SEM values, solid/dashed lines represent the model simulation with the best agreement to data, and the shaded areas represent the model uncertainty. (A), phosphorylation of HSL. (B), release of glycerol. (C), release of fatty acids (FA). (D), percentage of fatty acids being reesterified. Blue data/simulations correspond to normal conditions, and orange data/simulations correspond to type 2 diabetic conditions. The data for diabetic conditions in (A-C) and the data for normal conditions (D) were not used to train the model. In all panels, horizontal grey bars indicate where stimulation with isoproterenol (10 nM) was given. Furthermore, increasing concentrations of insulin were given together with 10nM isoproterenol in all points except one in all panels. The point without isoproterenol got no stimulus and is shown to the right in the graphs. The agreement with the rest of the original dataset (*in vitro* experiments) is shown in S4 Fig.

the model parameters. We can now use the model to not only predict the release of fatty acids *in vivo* in response to e.g. treatment with adrenaline, we can also predict the fatty acid release *in vivo* under diabetic conditions. As expected, the release of fatty acids *in vivo* in response to adrenaline temporally mimics the release of glycerol (Fig 7, cf. Fig 3). Furthermore, in line

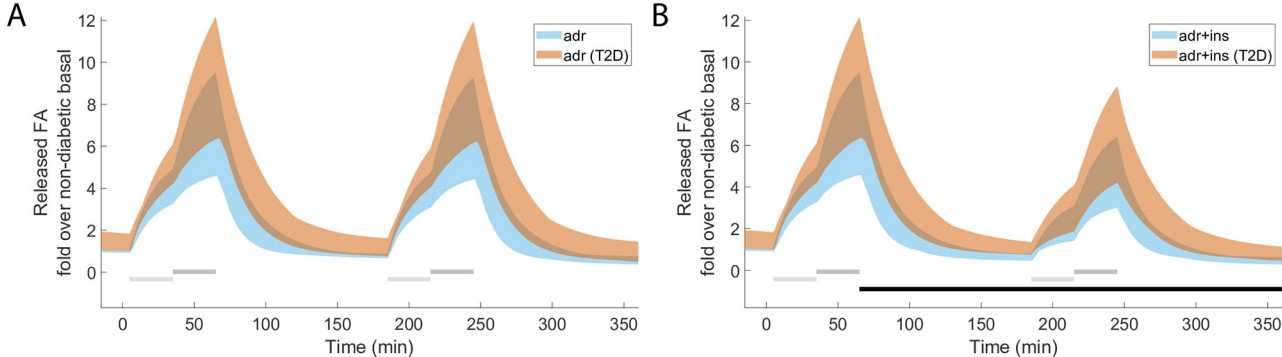

**Fig 7. Simulations of fatty acid release *in vivo*, under both non-diabetic (blue) and type 2 diabetic conditions (orange).** In both panels, the shaded areas represent the model simulation uncertainty. In both panels (A-B) horizontal light grey bars indicate where stimulation with 1 μM adrenaline was given, dark grey bars indicate where stimulation with 10 μM adrenaline was given, and horizontal black bars indicate where stimulation with 0.6 nM insulin was given. (A), model prediction without insulin, and (B), with insulin.

with the finding that the reesterification is impaired under the diabetic condition, resulting in an increased release of fatty acids from the adipocytes, the model predicts that the adipose tissue release of fatty acids *in vivo* is increased under diabetic conditions. Note that our model does not include mechanisms for fatty acid clearance due to uptake by other organs and therefore cannot predict the systemic levels of fatty acids or possible changes to the clearance in type 2 diabetes.

## Discussion

We examined the role of adipose tissue in the storage and release of fuel in the form of fat under normal conditions and when disturbed by insulin resistance and type 2 diabetes. To that end we present a mathematical model of hormonal control of lipolysis in human adipose tissue (Fig 1), a model that links molecular events at the cellular level with corresponding responses at the tissue level. The model can explain both *in vitro* dose-response data from isolated adipocytes (Figs 3D–3F and 6, and S4D–S4F Fig) and *in vivo* temporal data from microdialysis experiments in adipose tissue (Fig 3A–3C and S4A–S4C Fig), as well as accurately predict independent validation data (Fig 4).

There exist other models of lipolysis in humans. For example [28] modelled insulin levels and fatty acid release in response to glucose intake on a systemic level, but do not model any adrenergic stimulus or have a detailed intracellular compartment. A more extensive model of lipid metabolism [25], is also missing a detailed intracellular compartment and adrenergic stimulus. Conversely, [29] modelled lipolysis in response to adrenergic stimulus with a detailed intracellular compartment, but lacked insulin signalling. Tangentially, there also exist models of glucose homeostasis with insulin signalling, but lacking both lipolysis and adrenergic hormonal control [19, 30]. The model of hormonal control of lipolysis presented here is the first, to our knowledge, that includes insulin and adrenergic signalling, as well as an intracellular compartment. Some of the existing models are more detailed for certain aspects of lipolysis. We have chosen to only include sub-systems directly supported by experimental data, and the presented model can therefore be considered "minimal".

The presented model is in agreement with the experimental data (Figs 3 and 6, and S4 Fig) [15, 16]. Furthermore, the estimated uncertainty of the model is reasonably large compared to data uncertainty. This indicates that we have been successful in estimating the uncertainties of the model parameters and simulations.

Insulin action-3—the anti-lipolytic effect of insulin via α-adrenergic receptors—was needed to explain the combined effect of adrenaline, insulin, and phentolamine seen in the *in vivo* estimation data, as becomes clear when the action was removed (Fig 5D, notably in the second peak). It should be noted that insulin action-3, via the α-receptor, is also only observed at high concentrations of insulin, and may therefore be a secondary effect elicited by insulin in other cells or tissues.

In addition to investigating the contributions of the different insulin actions, we used the model to examine changes under type 2 diabetic conditions. The model shows that the extent of reesterification is altered from 66.7—74.3% under normal conditions to 39.6—64.1% under type 2 diabetic conditions. We also used the model to predict the temporal release of fatty acids *in vivo* in response to adrenaline in both non-diabetic and type 2 diabetic conditions (Fig 7). Type 2 diabetes have traditionally been associated with elevated levels of circulating fatty acids, an issue both challenged and affirmed [23, 31]. Our model predicts an *in vivo* increase in the release of fatty acids in diabetic conditions versus in non-diabetic conditions. It may appear surprising that evolution has provided us with the apparently wasteful reesterification of a substantial fraction of lipolytically released fatty acids. Nevertheless, the concept is far

from new and is extensively discussed in e.g. [15]. In effect, as fatty acids are fundamental to supply our energy requirements, they are also highly toxic and must always be strictly hindered from accumulating in cells. The reesterification is thus one aspect of the need to ensure that fatty acids do not accumulate, another aspect is the extremely intertwined regulatory pathways of lipolytic control.

We used data from two different sources [15, 16] to develop the model, which can be seen both as a strength and as a weakness. The use of internally consistent data, from the same laboratory under the same experimental conditions, is potentially important to test hypotheses and to unravel new biological mechanisms. For the purpose herein, to develop a first intracellular model of lipolysis that includes key observations and that later can be further built upon when more data become available, we believe it is a strength to use data from multiple sources. This means that the model is more general, and therefore more likely to be useful together with other human data from studies of adipose tissue lipolysis.

Desensitization, that cells decrease their response to continued or repeated stimuli, is a known phenomenon of β-adrenergic signalling. Stich et al. [16] controlled for desensitization by using multiple repeats of the stimulus paradigm. We decided to only include the first two rounds of stimulation of lipolysis, as we think there is a tendency to desensitization in the third stimulation, and we decided not to include this behaviour in this first model. There are other studies that show a clear desensitization in the release of glycerol in response to adrenaline [32, 33]. These studies have shorter intervals between stimuli (30 min, 1 h), and also reveal desensitization due to exercise-induced stimulation of lipolysis [33]. We have previously studied desensitization in heart cells using modelling and found that dose-response curves need to be adjusted for this phenomenon before important parameters such as the EC50 are computed [34]. Desensitization is an important phenomenon in β-adrenergic signalling that should not be overlooked and should be addressed in later models of lipolytic control.

Furthermore, in the *in vivo* data there were two different sets of data from the same experimental condition (adrenaline with insulin, Figs 2A and 3A in [16]). Since the model can only produce one simulation for identical conditions, we decided to not use two datasets for identical conditions to compute the cost. At the same time, we needed to scale the model states to be comparable to the experimental data. Therefore, we used the adrenaline with insulin data from Fig 2A in [16] to both compute optimal scaling parameters and the cost of the model, and only used the data from Fig 3A in [16] (shown in S1C Fig) to compute scaling parameters. We tested if any conclusions would change if we used the other dataset (from Fig 3A in [16]) to compute the cost, which it did not (see S1 Appendix).

During the modelling, we constrained insulin action-2 (Fig 1) so that the EC50 in response to insulin stayed between 0.5 and 1.1 nM. The reasoning behind this constraint is that known upstream signalling intermediaries, such as the autophosphorylation of the insulin receptor, has an EC50 in that range [17]. We also included a slightly delayed response to changes in adrenaline and isoproterenol stimulus *in vivo* when developing the model. Such a delay was observed in the microdialysis data [16], but not as obviously present in primary adipocytes [15]. In effect, we delayed the time for all changes of adrenaline and isoproterenol concentrations with 5 min when simulating the *in vivo* data (e.g. in Fig 3A–3C). The agreement between model simulations and data became substantially better with this delay. The underlying cause behind the observed delay in the microdialysis setup could be a delay in the microdialysis probe—or a biological tissue effect. We have chosen not to include the mechanisms of the delay in the model, instead we explicitly added the time delay.

The effect of insulin and adrenergic stimulation on the extent of reesterification in the diabetic condition is shown in Fig 6D. In short, the addition of adrenergic stimulation slightly lowers the reesterification. The single point without adrenergic stimulation (furthest to the

right) shows a slightly higher reesterification in the diabetic condition (yellow). On the other hand, increasing doses of insulin appears to have a negligible effect on the extent of reesterification. The estimated extent of reesterification for the diabetic condition (Fig 6D) is markedly lower than in the normal condition due to a higher overall release of fatty acids (Fig 6C), while the release of glycerol does not change (Fig 6B). Biologically, an increase in released fatty acids in the diabetic state can be seen as a way for adipocytes to get rid of excess fat to avoid an overload of fat.

In a sense our model is a model of an average adipocyte, where the outputs (glycerol and fatty acids) are scaled to the tissue scale (in vivo). At the same time, all in vitro experimental data are expressed as fold changes. We of course also scale the model simulations to be expressed as fold change during the comparison between the model and the in vitro data such that the simulations and data are expressed in the same way. Together this means that while the model can accurately explain the available biological data, the parts of the model where no biological data is available might be inaccurate. In a sense it is more relevant to see the model as having the right dynamics, and the work herein as a test of the structure of the model, rather than a fully detailed description of all states and scales.

An interesting aspect of our model is that it can be further developed in several directions. On the intracellular side, we aim to combine the model with our extensive work on the modelling of insulin signalling pathways in both non-diabetic and type 2 diabetic conditions [17–20]. With such a connection, we will reach a first comprehensive model for the human adipocyte that is based on extensive data from both non-diabetic and type 2 diabetic patients. This combined model will be able to simulate the major functions of the adipocyte: the control of lipolysis, as well as insulin control of glucose uptake, protein synthesis, and transcription. Further, on the systemic regulation of fatty acid release, the work herein opens up for a first connected model where intracellular components of lipolysis are connected to whole-body changes in fatty acid release. Such a connected model can also be combined with other models, for example models for intake of meals [21, 35–37]. Moreover, connected models are key to understand mechanisms of ectopic fat storage, i.e. where liver and muscle tissue increase their storage of lipids—a condition that is linked to disease development such as in type 2 diabetes, liver failure, and cardiovascular disease [38].

## Methods

### Data processing

*In vitro* experiments with isolated human adipocytes were performed by us and were previously published (Fig 14D in [15]). In the *in vitro* data, two points (for $10^{-7}$ and $10^{-6}$ M insulin in the non-diabetic fatty acid release data) had only two repeats, and we therefore set the data uncertainty for those two points to the average uncertainty of the non-diabetic fatty acid release. *In vivo* data from human microdialysis experiments were extracted from Figs 1A, 2A, and 3A in [16].

In all *in vivo* experiments [16], three sets of consecutive adrenergic stimuli were given. We have chosen to only include the first two sets of stimuli in the present study. The third set of stimuli was essentially a repeat of the second set of stimuli at a later time point—yet showed a different response than the second set of stimuli (not shown). The difference in the response can be technical and/or biological differences at this later time, differences not included in the model in the present study. Furthermore, we used only one of the two datasets with identical stimuli (adrenaline and insulin) for the calculation of the cost. However, we kept the other dataset when calculating the experimental scaling parameters.

## Mathematical modelling

A system of ordinary differential equations (ODEs) was used to model lipolysis and the release of fatty acids *in vitro* and *in vivo*. The equations are visualized in the interaction graph in Fig 1. The full model with all described equations can be found in a public repository (see **Data and model availability**). All ODEs are expressed in the time-scale of minutes, insulin is expressed in nM for the insulin actions.

## Equations for insulin actions

We modelled the three different actions of insulin described in [15, 16] (see Fig 1) as three separate sigmoidal functions, all dependent on the concentration of insulin. These three insulin functions affect downstream signalling proteins on PKB, PDE3B and the $\alpha_2$-adrenergic receptor. The equation for these sigmoidal functions is described in Eq (1).

$$Ins_x = 100 + \frac{min_x - 100}{1 + (ins/EC50_x)^{n_x}}, \qquad x \in \{1, 2, 3\} \tag{1}$$

$Ins_x$ is a function that is dependent on the given concentration of insulin ($ins$); $min_x$ is the minimum value the function has at ins = 0; $EC50_x$ is the concentration of insulin at which the function reaches half of the maximal response (which is set to 100). The steepness of the function is determined by $n_x$.

## Equations for insulin signalling

When IR is stimulated with insulin, a signalling cascade is initiated which leads up to the activation of PKB, through insulin receptor substrate 1 (IRS1), phosphoinositide 3-kinase (PI3K), and Phosphoinositide-dependent kinase-1 (PDK1). In our model, we have simplified this cascade as a direct action from insulin to the activation of PKB. PKB can also be activated by cAMP. We model PKB as either being in an inactive or an active configuration. The ODEs for PKB are given by Eq (2)

$$\begin{aligned} d/dt(PKB) &= -(k1a \cdot cAMP + k1a2 \cdot Ins_1) \cdot PKB + k1b \cdot PKBp \\ d/dt(PKBp) &= (k1a \cdot cAMP + k1a2 \cdot Ins_1) \cdot PKB - k1b \cdot PKBp \end{aligned} \tag{2}$$

Here, *PKB* and *PKBp* are the two model states for the inactive and active form of PKB, $Ins_1$ is the effect of insulin action-1, *cAMP* is the levels of cAMP, and *k1a k1a2*, and *k1b* are rate determining parameters.

Downstream, PKB will directly activate PDE3B. Beyond the direct activation by PKB, PDE3B will also be inactivated by insulin action-2. The detailed mechanism for this activation is currently unknown. We model PDE3B as being in either an inactive or an active state. The ODEs for PDE3B are given in Eq (3).

$$\begin{aligned} d/dt(PDE3B) &= -k2a \cdot PKBp \cdot PDE3B + k2b \cdot PDE3Ba \cdot Ins_2 \\ d/dt(PDE3Ba) &= k2a \cdot PKBp \cdot PDE3B - k2b \cdot PDE3Ba \cdot Ins_2 \end{aligned} \tag{3}$$

Here, *PDE3B* and *PDE3Ba* are the two model states for the inactive and active form of PDE3B, *PKBp* is the model state for activated PKB, $Ins_2$ is the effect of insulin action-2, and *k2a* and *k2b* are rate determining parameters.

## Equations for α₂-adrenergic receptor signalling

In the model, the α₂-adrenergic receptor can switch between two different configurations: activated or inactivated. This balance is offset towards the activated state when the α₂-adrenergic receptor is stimulated with adrenaline. The activated receptor will passively return to the inactive configuration. The activation of the α₂-adrenergic receptor is also augmented by insulin and inhibited by the addition of phentolamine. The ODEs for the α₂-adrenergic receptor are given in Eq (4).

$$d/dt(ALPHA) \quad = -(k3a \cdot Ins_3 \cdot adr + k3a2) \cdot (1 - phe\_effect \cdot phe) \cdot ALPHA + k3b \cdot ALPHAa$$
$$d/dt(ALPHAa) \quad = (k3a \cdot Ins_3 \cdot adr + k3a2) \cdot (1 - phe\_effect \cdot phe) \cdot ALPHA - k3b \cdot ALPHAa \tag{4}$$

Here, $ALPHA$ and $ALPHAa$ are the two model states for the inactive and active α₂-adrenergic receptor, $adr$ and $phe$ are the stimulation given as inputs, $Ins_3$ is insulin action-3, $k3a$, $k3a2$ and $k3b$ are rate determining parameters, and $phe\_effect$ is a parameter determining the effect of the phentolamine stimulation. In practice, $adr$ corresponds to the concentration of adrenaline (in nM), and $phe$ is a simplified boolean input (set to 1 if phentolamine is present, and 0 else).

## Equations for β-adrenergic receptor signalling

The β-adrenergic receptor, similarly to the α₂-adrenergic receptor, is also switching between two configurations. In contrast to the α₂-adrenergic receptor, the β-adrenergic receptor is also activated by adrenaline. Due to the uncertainty in difference in activation between adrenaline and isoproterenol, we added a scaling factor on isoproterenol. Furthermore, the activation of the β-adrenergic receptor is not increased by insulin or inhibited by phentolamine. The ODEs for the β-adrenergic receptor are given in Eq (5).

$$d/dt(BETA) \quad = -(k4a \cdot (iso \cdot isoscale + adr) + k4a2) \cdot BETA + k4b \cdot BETAa$$
$$d/dt(BETAa) \quad = (k4a \cdot (iso \cdot isoscale + adr) + k4a2) \cdot BETA - k4b \cdot BETAa \tag{5}$$

Here, $BETA$ and $BETAa$ are the two model states for the inactive and active β-adrenergic receptor, $adr$ and $iso$ are inputs corresponding to stimulation with adrenaline and isoproterenol, and $k4a$, $isoscale$, $k4a2$, and $k4b$ are rate determining parameters. $adr$ and $iso$ corresponds to the concentration (in nM) of adrenaline and isoproterenol respectively.

## Equations for lipolysis

Both the β-adrenergic receptor and the α₂-adrenergic receptor are G-protein coupled receptors. The G-proteins is made of multiple subunits, which will disassociate when the receptors are activated. These subunits will then go on and trigger another downstream effector. One such effector is adenylyl cyclase, which catalyse the conversion of ATP into cAMP. Through the G-proteins, β-adrenergic receptor will increase the activity of adenylyl cyclase, and α₂-adrenergic receptor will decrease the activity. In the model, we have simplified this interaction by ignoring the G-proteins, letting the β-adrenergic receptor and α₂-adrenergic receptor directly affect the adenylyl cyclase. Furthermore, we model the adenylyl cyclase as being either inactive or active, where the active version leads to an increased production of cAMP. The model equations are given in Eq (6).

$$d/dt(AC) \quad = -k5a \cdot (BETAa) \cdot AC + k5b \cdot (ALPHAa) \cdot ACa$$
$$d/dt(ACa) \quad = k5a \cdot (BETAa) \cdot AC - k5b \cdot (ALPHAa) \cdot ACa \tag{6}$$

Here, *AC* and *ACa* are the two model states for inactive and active adenylyl cyclase respectively, *BETAa* and *ALPHAa* are the model states for active β- and α-receptor, and *k5a* and *k5b* are rate determining parameters.

Downstream of both adenylyl cyclase and PDE3B is cAMP. An increase in adenylyl cyclase activation will lead to an increased concentration of cAMP, and an increase in PDE3B activation will lead to a decreased concentration of cAMP. Together, adenylyl cyclase and PDE3B balance the concentration of cAMP in the cell. cAMP have negative feedback loop by activating PKB via PI3K, which in turn activates PDE3B, which leads to a decreased concentration of cAMP. We have also added both a basal production and degradation of cAMP. The ODEs for cAMP are given in Eq (7).

$$d/dt(cAMP) = k6a \cdot ACa - k6b \cdot PDE3Ba \cdot cAMP \tag{7}$$

Here, *cAMP* is the model state for cAMP, *ACa* and *PDE3Ba* are the model states for the active configurations of adenylyl cyclase and PDE3B, and *k6a* and *k6b* are rate determining parameters.

The concentration of cAMP indirectly controls the lipolysis by activation of protein kinase A (PKA), which in turn will activate the lipid droplet-coating protein perilipin 1 (PLIN1) and hormone-sensitive lipase (HSL). Activation of PKA leads to the phosphorylation of the lipid droplet-coating protein perilipin 1 (PLIN1) and its subsequent release of the adipose triacylglycerol lipase (ATGL) activator, comparative gene identification-58. Active ATGL will catalyse the hydrolysis of the first fatty acid of the triacylglycerol. Phosphorylation of HSL by PKA results in activation and translocation of HSL to the lipid droplet, where HSL hydrolyses the second fatty acid leaving monoacylglycerol to be hydrolysed by a constitutively active monoacylglycerol lipase. HSL is capable to hydrolyse triacylglycerol, but ATGL is believed to be more important in this rate-limiting step of lipolysis. In the model, we simplified these interactions by only modelling HSL with an input from cAMP. The ODEs are given in Eq (8).

$$\begin{aligned} d/dt(HSL) &= -k7a \cdot cAMP \cdot HSL + k7b \cdot HSLp \\ d/dt(HSLp) &= k7a \cdot cAMP \cdot HSL - k7b \cdot HSLp \end{aligned} \tag{8}$$

Here, *HSL* and *HSLp* are the states for inactive and active HSL, *cAMP* is the model state for cAMP, and *k7a* and *k7b* are the rate determining parameters.

Once the triacylglycerol has been broken down into three fatty acids and one glycerol, the glycerol will be transported out of the cell, and the fatty acids will either be transported out or reesterified with glycerol-3P into new triacylglycerol. This reesterification is reduced in type 2 diabetes. Some of the fatty acids can also go back into the cell, while the glycerol cannot. Fatty acids and glycerol outside of the cell will be cleared in an *in vivo* setting, but not *in vitro*. The ODEs are given in Eq (9).

$$\begin{aligned} d/dt(Gly) &= k8a \cdot HSLp - k8b \cdot Gly \\ d/dt(FFA) &= 3 \cdot k8a \cdot HSLp - (k8b + k8c \cdot diab) \cdot FFA \end{aligned} \tag{9}$$

Here, *Gly* and *FFA* are the states for glycerol and fatty acids, *HSLp* is the state for activated HSL, *diab* is a parameter controlling the reduction of reesterification under diabetic conditions, and *k8a*, *k8b*, and *k8c* are rate determining parameters. *k8a* corresponds to the transport of fatty acids and glycerol from the inside to the outside of cell, *k8b* is the clearance of fatty acids and glycerol *in vivo* (clearance was disabled *in vitro* by setting *k8b* = 0), and *k8c* is the reesterification of fatty acids into triacylglycerol. To simulate the effect of type 2 diabetes on the reesterification, the parameter *diab* was allowed to vary between 0.0—1.0. In non-diabetic

conditions, the type 2 diabetes effect was disabled by setting $diab = 1$ (i.e. no effect of type 2 diabetes).

## Translating the model states to experimental data

We constructed measurement equation to translate the model states of our model to the corresponding *in vivo* experimental data. In practice, we introduced a linear drift, a scaling constant and an offset constant. The measurement equation for glycerol is illustrated in Eq (10).

$$\hat{y}_{Gly} = k_{scale} \cdot (Gly - k_{drift} \cdot time) + k_{offset} \tag{10}$$

Here, $Gly$ and $\hat{y}_{Gly}$ are the model state and measurement equation for glycerol, $k_{drift} \cdot time$ is the drift over time, $k_{scale}$ is the scaling constant and $k_{offset}$ is the offset constant. The scaling and offset constants were calculated using MATLABs least squares with known covariance (lscov) function.

For the *in vitro* experiments we did not use a measurement equation, but we did scale the simulations to be "fold over iso stimulation only", as was done in the experimental data.

## Initial values

All states corresponding to activations were represented as per cent of activation, i.e. the sum of the two states will be 100. All initial values of the ODEs were set to arbitrary non-negative values:

$BETA(0) = 80$, $BETAa(0) = 20$, $ALPHA(0) = 80$, $ALPHAa(0) = 20$, $AC(0) = 80$,

$ACa(0) = 20$, $PKB(0) = 80$, $PKBp(0) = 20$, $PDE3B(0) = 80$, $PDE3Ba(0) = 20$,

$cAMP(0) = 0$, $HSL(0) = 80$, $HSLp(0) = 20$, $Gly(0) = 0$, $FFA(0) = 0$

We then simulated the model without any stimuli to numerically calculate the steady state, which was used as initial values for the simulations of the experiments with stimuli.

## Calculating the percentage of reesterification

We calculate the percentage of reesterification using Eq (11), the same way as the calculation was done for the experimental data in [15].

$$100 \cdot \frac{3 \cdot glycerol - FFA}{3 \cdot glycerol} \tag{11}$$

## Quantifying the model agreement to experimental data

In order to evaluate the model's performance, we quantified the model agreement to data using a function typically referred to as a cost function. In this case, we used the normalized sum of squared residual as cost function (Eq (12)).

$$v(\theta) = \sum_t \left( \frac{y_t - \hat{y}_t(\theta)}{SEM_t} \right)^2 \tag{12}$$

Here, $v(\theta)$ is the cost, equal to the sum of the normalized residual over all measured time points, $t$; $p$ is the parameters; $y_t$ is the measured data and $\hat{y}_t(\theta)$ is the model simulations; $SEM_t$ is the standard error of the mean for the measured data.

## Statistical analysis

To reject models, we used the $\chi^2$-test with a significance level of 0.05. We used 137 degrees of freedom for the original training data (144 data points, minus 7 scaling parameters) leading to a threshold for rejection of $\chi^2(0.05, 137) \approx 165.3$. For the extended set of experimental data, used after diabetes was introduced (from the section **Estimating the extent of altered reesterification in type 2 diabetes** in the results), we used 152 degrees of freedom, resulting in a threshold for rejection of $\chi^2(0.05, 152) \approx 181.8$. Any combination of model and parameter set that results in a cost (Eq (12)) above the threshold must be rejected. If no parameter set exists for a model that results in a sufficiently low cost, the model structure must be rejected.

## Uncertainty estimation

The uncertainty of both the parameters and the model simulations for estimation, validation, and predictions, were gathered as proposed in [39] and implemented in [27]. In short, we estimate the model uncertainty by subdividing the problem into multiple optimization problems, with one problem per model property ($\hat{p}$) of interest. Here, the property $\hat{p}$ corresponds to either a parameter value or a simulation at a specific time. We solved a subproblem by either minimizing or maximizing the value of the property by tuning the values of all model parameter while requiring the cost (Eq (12)) to be below the $\chi^2$-threshold. By finding the minimal and maximal value of the property ($\hat{p}_{min}$, and $\hat{p}_{max}$ respectively), we get a bound on the uncertainty ($\hat{p}_{min} - \hat{p}_{max}$). This approach is similar to what is done in traditional profile-likelihood analysis, where a property $\hat{p}$ is fixed at some value and the cost is minimized. The property is then decreased iteratively as long as the cost is below the $\chi^2$-threshold (i.e., $cost < \chi^2 - threshold$). In other words, the minimal value ($\hat{p}_{min}$) is found when the cost reaches the $\chi^2$-threshold. This process is then repeated by now *increasing* the property iteratively to find the maximal value ($\hat{p}_{max}$). The traditional profile-likelihood analysis problem is typically formulated as in Eq (13):

$$\begin{aligned} \text{minimize} \quad & v(\theta) \\ \text{sunject to} \quad & \hat{p} = p. \end{aligned} \tag{13}$$

where the cost $v(\theta)$ is minimized while the property $\hat{p}$ is fixed to a value $p$. The value $p$ is decreased (and later increased) iteratively to find the boundaries of the property. Here, we inverse the problem and solve it directly, rather than step through the values of the property. The formulation for this inverse problem is given in Eq (14):

$$\text{minimize} \quad \hat{p} \tag{14a}$$

$$\text{sunject to} \quad v(\theta) \leq \chi^2. \tag{14b}$$

where $\hat{p}$ is minimized to find the lower bound on the value of the property, while requiring the cost $v(\theta)$ to be below the $\chi^2$-threshold. To get the upper bound, the problem in Eq (14) can be solved as a maximization problem. In practice, the constraint (Eq (14b)) is relaxed into the objective function as a L1 penalty term with an offset if $v(\theta) > \chi^2$.

$$\text{minimize} \quad \hat{p} + penalty \tag{15a}$$

$$\text{sunject to} \quad penalty = \begin{cases} (1 + |\hat{p}|) \cdot (1 + |V(\theta) - \chi^2|), & \text{if } V(\theta) > \chi^2 \\ 0, & \text{otherwise} \end{cases} \tag{15b}$$

To find the upper bound on the uncertainty, we solved the maximization problem as a minimization problem (Eq (15)) by changing sign of the property in the objective function to $-\hat{p}$.

## Optimization and software

We used MATLAB R2020a (MathWorks, Natick, MA) and the IQM toolbox (IntiQuan GmbH, Basel, Switzerland), a continuation of [40], for modelling. The parameter values were estimated using the enhanced scatter search (eSS) algorithm from the MEIGO toolbox [41]. eSS were restarted multiple times, partially run in parallel at the local node of the Swedish national supercomputing centre (NSC). We allowed the parameter estimation to freely find the best possible combinations of parameter values, within boundaries. The bounds of the parameter values are given in S2 Table.

## Data and model availability

The experimental data as well as the complete code for data analysis and modelling are available at https://github.com/willov/lipolysis (DOI: 10.5281/zenodo.5639332) and is mirrored at https://gitlab.liu.se/ISBgroup/projects/lipolysis.

## Supporting information

**S1 Table. All bounds and estimated values for the free parameters.** The parameters were allowed to vary in the range given in S2 Table. For the specific parameter being investigated, the bound was relaxed and the threshold for when a parameter was deemed nonidentifiable was set to the value given in the table in columns Lower threshold and Upper threshold). The minimum and maximal found values of a parameter is given in columns $\theta_{original}^{min}$, and $\theta_{original}^{max}$ respectively.
(PDF)

**S2 Table. Bounds used for optimization of the free parameters, and the sets of optimal values.** The rate parameters ($kx$) were given a free range ($10^{-6}$ to $10^{6}$). *isoscale* was allowed a 20% deviation from the expected value of 10. For the input functions, the minimum values $min_x$ was given a range from zero to 20% of max, the steepness $n_x$ was given a range from 0 to 2, and the $EC50_x$ was given a free range for all doses used in the dataset from [15] ($10^{-5}$ to $10^{3}$ nM), except for $EC50_1$ which was limited based on the EC50 of IR in [17]. $\theta_{original}^*$ corresponds to the optimal parameter set for the original dataset. $\theta_{extended}^*$ corresponds to the optimal parameter set for the extended dataset.
(PDF)

**S1 Fig. Model agreement with experimental data, with overlaid *in vivo* experiments.** In all panels, solid lines represent the model simulation with the best agreement to data, the shaded areas represent the model uncertainty, and experimental data point are represented as mean values with error bars (SEM). (A-C), *in vivo* time-series experiments. (D-F), *in vitro* dose-response experiments. In all subfigures, horizontal bars indicate where stimulations were given. In (A-C), light/dark grey bars indicate low/high adrenergic stimulus (1/10 μM adrenaline or 0.1/1 μM isoproterenol) with or without phentolamine (phe; 100 μM), black bars indicate stimulation with insulin (0.6 nM). In (D-F) grey bars indicate stimulation with isoproterenol (10 nM). In the *in vivo* experiments, experiments with adrenaline are shown in light blue (A-C), with isoproterenol in purple (B), and with the combined stimulation with adrenaline and phentolamine in green (C). In the *in vitro* experiments (D-F), increasing doses of insulin were given together with 10nM isoproterenol in all points except one. The point without isoproterenol got no stimulus and is shown to the right in the graphs.
(PDF)

**S2 Fig. Prediction of intracellular extent of HSL phosphorylation.** The shaded area represents the model uncertainty, and the experimental data are represented as mean values with error bars (SEM). The horizontal grey bar indicates where stimulation with isoproterenol (10 nM) have been given. Increasing doses of insulin were given together with 10 nM isoproterenol in all points except one. The point without isoproterenol got no stimulus and is shown to the right in the graph.
(PDF)

**S3 Fig. Model agreement with experimental data, with overlaid *in vivo* experiments, for the model without insulin action-3.** In all panels, solid lines represent the model simulation with the best agreement to data, the shaded areas represent the model uncertainty, and experimental data point are represented as mean values with error bars (SEM). (A-C), *in vivo* time-series experiments. (D-F), the *in vitro* dose-response experiments. In all subfigures, horizontal bars indicate where stimulations were given. In (A-C), light/dark grey bars indicate low/high adrenergic stimulus (1/10 μM adrenaline or 0.1/1 μM isoproterenol) with or without phentolamine (phe; 100 μM), black bars indicate stimulation with insulin (0.6 nM). In (D-F) grey bars indicate stimulation with isoproterenol (10 nM). In the *in vivo* experiments, experiments with adrenaline are shown in light blue (A-C), with isoproterenol in purple (B), and with the combined stimulation with adrenaline and phentolamine in green (C). In the *in vitro* experiments (D-F), increasing doses of insulin were given together with 10nM isoproterenol in all points except one. The point without isoproterenol got no stimulus and is shown to the right in the graphs.
(PDF)

**S4 Fig. Model agreement for the experiments not shown in Fig 4.** Estimation data and model simulations, for the original data (e.g. used in Fig 3). In all panels, solid lines represent the model simulation with the best agreement to data, and experimental data point are represented as mean values with error bars (SEM). (A-D), *in vivo* time-series experiments. (E-G), *in vitro* dose-response experiments. In all subfigures, horizontal bars indicate where stimulations were given. In detail, light/dark grey bars indicate stimulation with: 1/10 μM adrenaline in (A, B), 0.1/1 μM isoproterenol in (C), and 1/10 μM adrenaline with 100 μM phentolamine. Black bars in (B-D) indicates stimulation with 0.6 nM insulin. In (E-G) grey bars indicate stimulation with isoproterenol (10 nM). In the *in vivo* experiments, experiments with adrenaline are shown in light blue (A-C), with isoproterenol in purple (B), and with the combined stimulation with adrenaline and phentolamine in green (C). In the *in vitro* experiments (D-F), increasing doses of insulin were given together with 10 nM isoproterenol in all points except one. The point without isoproterenol got no stimulus and is shown to the right in the graphs. An alternative visualization is available in S5 Fig showing the difference by overlaying the experiments.
(PDF)

**S5 Fig. Model agreement for the experiments not shown in Fig 4, with overlaid *in vivo* experiments.** Estimation data and model simulations, for the original data (e.g. used in Fig 3). In all panels, solid lines represent the model simulation with the best agreement to data, the shaded areas represent the model uncertainty, and experimental data point are represented as mean values with error bars (SEM). (A-C), *in vivo* time-series experiments. (D-F), *in vitro* dose-response experiments. In all subfigures, horizontal bars indicate where stimulations were given. In (A-C), light/dark grey bars indicate low/high adrenergic stimulus (1/10 μM adrenaline or 0.1/1 μM isoproterenol) with or without phentolamine (phe; 100 μM), black bars indicate stimulation with insulin (0.6 nM). In (D-F) grey bars indicate stimulation with isoproterenol (10 nM). In the *in vivo* experiments, experiments with adrenaline are shown in light

blue (A-C), with isoproterenol in purple (B), and with the combined stimulation with adrenaline and phentolamine in green (C). In the *in vitro* experiments (D-F), increasing doses of insulin were given together with 10nM isoproterenol in all points except one. The point without isoproterenol got no stimulus and is shown to the right in the graphs.
(PDF)

**S1 Appendix.**
(PDF)

## Author Contributions

**Conceptualization:** Peter Strålfors, Gunnar Cedersund, Elin Nyman.

**Data curation:** Cecilia Jönsson, Peter Strålfors, Elin Nyman.

**Formal analysis:** William Lövfors.

**Funding acquisition:** Peter Strålfors, Gunnar Cedersund, Elin Nyman.

**Investigation:** William Lövfors, Jona Ekström, Cecilia Jönsson, Elin Nyman.

**Methodology:** William Lövfors, Elin Nyman.

**Resources:** Peter Strålfors, Gunnar Cedersund, Elin Nyman.

**Software:** William Lövfors.

**Supervision:** Gunnar Cedersund, Elin Nyman.

**Validation:** William Lövfors, Elin Nyman.

**Visualization:** William Lövfors, Elin Nyman.

**Writing – original draft:** William Lövfors, Cecilia Jönsson, Peter Strålfors, Gunnar Cedersund, Elin Nyman.

**Writing – review & editing:** William Lövfors, Jona Ekström, Cecilia Jönsson, Peter Strålfors, Gunnar Cedersund, Elin Nyman.

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
