## [Decision Letter · Decision Letter 0]

27 Sep 2021

PONE-D-21-26477A systems biology analysis of lipolysis and fatty acid release from adipocytes *in vitro* and from adipose tissue *in vivo*PLOS ONE

Dear Dr. Lövfors,

Thank you for submitting your manuscript to PLOS ONE. After careful consideration, we feel that it has merit but does not fully meet PLOS ONE’s publication criteria as it currently stands. Therefore, we invite you to submit a revised version of the manuscript that addresses the points raised during the review process. While the biological data had been published before and appear suitable for a first minimal model, the mathematical modelling/statistacal analysis and specific selection of biological data-points need to be addressed very carefully in the revision process.

We look forward to receiving your revised manuscript.

Kind regards,

Monika Oberer

Academic Editor

PLOS ONE

Journal Requirements:

Reviewers' comments:

Reviewer's Responses to Questions

**Comments to the Author**

1. Is the manuscript technically sound, and do the data support the conclusions?

Reviewer #1: Yes

Reviewer #2: Yes

Reviewer #3: Yes

2. Has the statistical analysis been performed appropriately and rigorously? 

Reviewer #1: I Don't Know

Reviewer #2: Yes

Reviewer #3: I Don't Know

3. Have the authors made all data underlying the findings in their manuscript fully available?

Reviewer #1: Yes

Reviewer #2: Yes

Reviewer #3: Yes

4. Is the manuscript presented in an intelligible fashion and written in standard English?

Reviewer #1: Yes

Reviewer #2: Yes

Reviewer #3: No

5. Review Comments to the Author

Reviewer #1: In this study, Lövfors et al. developed a minimal mechanistic model to assess the impact of hormonal control of human adipocyte intracellular triglyceride hydrolysis (lipolysis). Several hormones regulate human adipocyte lipolysis: (Nor)epinephrine act via alpha2- and beta1/beta2-adrenergic receptors (AR) while the functional role of beta3-AR in humans is controversial; and insulin acts through the insulin receptor (IR). Between these different signaling pathways, a high degree of crosstalk exists that depends on hormone concentration. Thus, this new model shall enable to predict lipolysis during different metabolic conditions.

This is a very important topic as lipolysis is dysregulated in metabolic diseases such as obesity and type 2 diabetes and mathematic models will help to develop potential new pharmaceutical targets.

To feed their mathematic models, the authors have used available data from two different already published studies: one of their own publication (doi:10.1042/BCJ20190594) and one from Stich et al. (doi:10.1152/ajpendo.00502.2002). The data sets comprise in-vitro and in-vivo experiments using isolated mature adipocytes from subcutaneous adipose tissue and microdialysis data sets - both from non-diabetic individuals.

In vitro, lipolysis was stimulated for 10 min using 10 nM of the beta-AR agonist isoproterenol (ISO) and inhibited by varying insulin concentrations. In situ, lipolysis was stimulated by either ISO (0.1 and 1 µM) or epinephrine (1 and 10 µM) and inhibited by insulin or phentolamine (a nonselective alpha-AR antagonist) in 8 healthy subjects. Two cylces of repeated epinephrine perfusions had been used to feed the model instead of three that had been conducted in order to exclude possible effects of receptor desensitization. Fatty acid (FA) and glycerol released into the media and in the extracellular fluid were determined as a measure of intracellular lipolysis. In addition, immunoblot analyses of the phosphorylation status of two important proteins involved in intracellular lipolysis, namely hormone-sensitive lipase and protein kinase B, have been included in the in-vitro model. However, and this owes to the published data, information on other parameters that are crucial for lipolysis such as adipose triglyceride lipase, alpha/beta hydrolase domain containing protein 5, IR are missing.

The model included three different actions of the insulin: i) Inhibition of lipolysis via protein kinase B and phosphodiesterase 3B (PDE3B); ii) Stimulation of lipolysis via inhibition of PDE3B; and iii) Inhibition of lipolysis via alpha-AR. The experimental settings are comprehensible and have been previously published. Thus, the two studies are reasonable to be used for mathematic modeling.

My expertise does not lay in the field of mathematic modeling. Thus, I cannot comment on neither the modeling part nor the applied statistical tests. The authors have sufficiently referenced the literature and all findings are available in the main or supplemental text of the ms.

Overall, the manuscript is very well written in standard English without any obvious typos or grammatical errors. The graphs are clearly presented and described. However, I suggest that the authors are more precise on the title that fits to the study.

Reviewer #2: I have read this manuscript with great care and interest. The authors present a mathematical model that links in vitro and in vivo information on the action of insulin in the lipolysis mechanism of patients with type 2 diabetes mellitus. In particular, the work presented offers insight into the high circulation of fatty acids in patients under this pathological condition. The integrated model of in vitro and in vivo data provides an insight into the mechanisms of ectopic fat storage.

I consider the results to be scientifically interesting, mainly for a broad computational and/or theoretical biology community. The authors present a mathematical model of hor- monal control of lipolysis in human adipose tissue, linking molecular events at the cellular level with corresponding responses at the tissue level. Although the procedures and results of the article are potentially suitable for a PLOS ONE journal; I must encourage the authors to make a thorough review of the submitted manuscript, taking carefully into account the following points.

Further review is attached as a pdf file.

Reviewer #3: The manuscript concerns a system biological analysis of a "minimal" mathematical model of the regulation and release of fatty acid via lipolysis. In general the peer is interesting and seems quite sound. However, even after reading all parts (partly several time)

there are still way too many aspect which remain unclear. The manuscript is clearly not ready for publication and should be carefully reworked before it could be suitable for publication.

In the following, I am only state some major examples and questions, which need to be addressed.

The process of data selection and parameter identification is unclear. For example that authors mention [4] as a source of measured data in the beginning, but not during the discussion of the Figs 3 ff. Moreover they only mention in the discussion why the data from [4] are reduced to the first two peaks. The proceedings leading to Tables S2 and S1 and their relationship is not made understandable. The author state that non-identifiable parameters are removed but not if these are parameters of the presented model or otherwise which processes have been eliminated from the model. They state that they brake their own rule by keeping parameter k8c, which should be discussed and argued better. The description of the uncertainty estimation is not clear, the property p hat not explained and it remains unclear in which sense it can be minimised. The authors do not address that removing insulin action 3 would pass their own chi^2 criteria of admissible parameters sufficiently. The model itself involves heuristic terms, sometimes nonlinear sometimes linear, a mixture which is sufficiently arbitrary, but can be imagined as a minimal model. However only one (not very good) example of data verification gives -- together with the above mentioned inconsistencies -- the feel that be model might be useful, but that its argumentation is weak.

Some more comments: Looking at the Fig. 4 and the parameters, one could speculate if the model for lipolysis is not good enough.

Basically the models assumes all hydrolysis steps saturated. Is PKBp equal PKB_A in Fig 1? The authors state that only one of the two data sets of [4] is used. What happens if the other one is used initially? The authors should comment on the 2/3 reesterification rate in healthy humans. Is it conceivable that evolution has produce a mechanism where 2/3/ of the produced FA are not released but immediately reesterfied?

6. PLOS authors have the option to publish the peer review history of their article (what does this mean?). If published, this will include your full peer review and any attached files.

Reviewer #1: No

Reviewer #2: No

Reviewer #3: No

---

## [Author Response · Author response to Decision Letter 0]

8 Nov 2021

We are thankful for the reviewers comments. We have addressed the comments in the attached document 'Response to Reviewers'.

---

## [Decision Letter · Decision Letter 1]

1 Dec 2021

PONE-D-21-26477R1A systems biology analysis of lipolysis and fatty acid release from adipocytes *in vitro* and from adipose tissue *in vivo*PLOS ONE

Dear Dr. Lövfors,

Thank you for submitting your manuscript to PLOS ONE. After careful consideration, we feel that it has merit but does not fully meet PLOS ONE’s publication criteria as it currently stands. Therefore, we invite you to submit a revised version of the manuscript that addresses the points raised during the review process. There are only minor revision left which, if properly addressed, do not go out into another reviewing round before final acceptance.

We look forward to receiving your revised manuscript.

Kind regards,

Monika Oberer

Academic Editor

PLOS ONE

Journal Requirements:

Reviewers' comments:

Reviewer's Responses to Questions

**Comments to the Author**

1. If the authors have adequately addressed your comments raised in a previous round of review and you feel that this manuscript is now acceptable for publication, you may indicate that here to bypass the “Comments to the Author” section, enter your conflict of interest statement in the “Confidential to Editor” section, and submit your "Accept" recommendation.

Reviewer #1: All comments have been addressed

Reviewer #2: All comments have been addressed

Reviewer #3: (No Response)

2. Is the manuscript technically sound, and do the data support the conclusions?

Reviewer #1: Yes

Reviewer #2: Yes

Reviewer #3: Yes

3. Has the statistical analysis been performed appropriately and rigorously? 

Reviewer #1: I Don't Know

Reviewer #2: Yes

Reviewer #3: Yes

4. Have the authors made all data underlying the findings in their manuscript fully available?

Reviewer #1: Yes

Reviewer #2: Yes

Reviewer #3: Yes

5. Is the manuscript presented in an intelligible fashion and written in standard English?

Reviewer #1: Yes

Reviewer #2: Yes

Reviewer #3: Yes

6. Review Comments to the Author

Reviewer #1: I have no further comments and thus congratulate the authors to their piece of work.

Reviewer #2: I thank the authors for the responses provided to each previously made statement. Next, I list new comments to these answers, if I consider them necessary:

1. No additional comments.

2. No additional comments.

3. An apology for the omission of the year of publication of the previously suggested reference. I mean the work of

López-Palau, N. E. et al. "Mathematical model of blood glucose dynamics by emulating the pathophysiology of glucose metabolism in type 2 diabetes mellitus." Scientific Reports 10.1 (2020): 1-11.

Precisely, this article discusses a generalized compartmental model based on the analysis of dynamic mass conservation systems under the implicit idea of the virtual patient for the comparison of the model with clinical data. An exciting part of their work is that this model can be extended, according to the authors, to include fatty acid dynamics. The model works on the same length and time scales as clarified in point 1.

4. As the authors say, this can be a piece to build a short report independent of the submitted one.

5. Additional review. No additional comments.

It only remains for me to thank the authors again for their professionalism in preparing this article, which is undoubtedly a research product of the highest quality. With the slight corrections you can make to my new suggestions, I consider this article appropriate to be published in this Journal.

Reviewer #3: Overall the manuscript was improved and seems now (almost) fit for publication.

I appreciate the alternative analysis in Appendix S8 as well as the improved discussions.

I am still unsure about what exactly the authors mean on page 4 with (downwards) non-identifiable,

since what is or is not identifiable does interlink so strong with the methods.

I would appreciate that to be made clearer.

In the uncertainty estimation, it is still unclear if the property p is a scalar, i.e. the minimisation

was done one parameter a time with the others fixed or multidimensional. That should be made clear.

7. PLOS authors have the option to publish the peer review history of their article (what does this mean?). If published, this will include your full peer review and any attached files.

Reviewer #1: No

Reviewer #2: No

Reviewer #3: No

---

## [Author Response · Author response to Decision Letter 1]

6 Dec 2021

We have responded to all the reviewer comments in the "Response to Reviewers" document. 

We have also supplied our updated financial statement in the "Cover letter" document.

---

## [Editor Report · Decision Letter 2]

9 Dec 2021

A systems biology analysis of lipolysis and fatty acid release from adipocytes *in vitro* and from adipose tissue *in vivo*

PONE-D-21-26477R2

Dear Dr. Lövfors,

We’re pleased to inform you that your manuscript has been judged scientifically suitable for publication and will be formally accepted for publication once it meets all outstanding technical requirements.

Kind regards,

Monika Oberer

Academic Editor

PLOS ONE
---

## [Editor Report · Acceptance letter]

21 Dec 2021

PONE-D-21-26477R2 

A systems biology analysis of lipolysis and fatty acid release from adipocytes *in vitro* and from adipose tissue *in vivo*

Dear Dr. Lövfors:

I'm pleased to inform you that your manuscript has been deemed suitable for publication in PLOS ONE. Congratulations! Your manuscript is now with our production department. 

Kind regards, 

on behalf of

Dr. Monika Oberer 

Academic Editor

PLOS ONE